# Retrogressive thaw slumps along the Qinghai-Tibet Engineering Corridor: a comprehensive inventory and their distribution characteristics

Zhuoxuan Xia[1], Lingcao Huang[1†], Chengyan Fan[2], Shichao Jia[2], Zhanjun Lin[3], Lin Liu[1], Jing Luo[3],
Fujun Niu[3], and Tingjun Zhang[2] (deceased)

[1] Earth System Science Programme, Faculty of Science, The Chinese University of Hong Kong, Hong Kong SAR, China
[2] Key Laboratory of West China's Environments (DOE), College of Earth and Environmental Sciences, Lanzhou University, Lanzhou, China
[3] Northwest Institute of Eco-Environment and Resources, Chinese Academy of Sciences, Lanzhou, China

[†] Now at Earth Science and Observation Center, Cooperative Institute for Research in Environmental Sciences, University of Colorado Boulder, Boulder, CO, USA

*Correspondence to*: Lingcao Huang (huanglingcao@link.cuhk.edu.hk)

**Abstract.** The important Qinghai Tibet Engineering Corridor (QTEC) covers the part of the Highway and Railway underlain by permafrost. The permafrost on the QTEC is sensitive to climate warming and human disturbance and suffers accelerating degradation. Retrogressive thaw slumps (RTSs) are slope failures due to the thawing of ice-rich permafrost. They typically retreat and expand at high rates, damaging infrastructure, and releasing carbon preserved in frozen ground. Along the critical and essential corridor, RTSs are commonly distributed but remain poorly investigated. To compile the first comprehensive inventory of RTSs, this study uses an iteratively semi-automatic method built on deep learning to delineate thaw slumps in the 2019 PlanetScope CubeSat images over a ~54,000 km$^2$ corridor area. The method effectively assesses every image pixel using DeepLabv3+ with limited training samples and manually inspects the deep-learning-identified thaw slumps based on their geomorphic features and temporal changes. The inventory includes 875 RTSs, of which 474 are clustered in the Beiluhe region, and 38 are near roads or railway lines. The dataset is available at https://doi.org/10.5281/zenodo.6397028 (Xia et al., 2021), with the Chinese version at DOI: 10.11888/Cryos.tpdc.272672. These RTSs tend to be located on north-facing slopes with gradients of 1.2°–18.1° and distributed at medium elevations ranging from 4511 to 5212 m a.s.l. They prefer to develop on land receiving relatively low annual solar radiation (from 2900 to 3200 kWh m$^{-2}$), alpine meadow covered, and loam underlay. Our results provide a significant and fundamental benchmark dataset for quantifying thaw slump changes in this vulnerable region undergoing strong climatic warming and extensive human activities.

**Short Summary**

Retrogressive thaw slumps are slope failures resulting from abrupt permafrost thaw, widely distributed along the Qinghai-Tibet Engineering Corridor. The potential damage to infrastructure and the carbon emission of thaw slumps motivated us to

obtain an inventory of thaw slumps. We used a semi-automatic method to map 875 thaw slumps, filling the knowledge gap of thaw slump locations and providing key benchmarks for analyzing the distribution features and quantifying spatio-temporal changes.

## 1    Introduction

Permafrost is defined as ground that remains at or below 0℃ for at least two consecutive years (Van Everdingen, 1998; French, 2018). On the Qinghai-Tibet Plateau, permafrost covers an area of about $1.06 \times 10^6$ km$^2$ (Zou et al., 2017; Cao et al., 2019) with an average elevation of more than 4000 m (Liu, 2000) and latitudes of 26°N–38°N (Wang and French, 1994; Zhang et al., 2008). Because the underlying permafrost on the plateau is characterized by shallow thickness and relatively high temperature (Ran et al., 2022; Wu and Zhang, 2008; Wu et al., 2010; Zhao et al., 2021; Zhou et al., 2000), it is vulnerable to

degradation under climate warming and disturbance due to human activities. One critical zone suffering accelerated permafrost degradation is the Qinghai-Tibet Engineering Corridor (QTEC), which contains the Qinghai-Tibet Railway and Qinghai-Tibet Highway. This corridor is 1120 km long, and almost half its length (531 km) is underlain by permafrost (Jin et al., 2008; Wu and Zhang, 2010).

As a typical type of thermokarst landform, retrogressive thaw slumps (RTSs) are caused by the thawing of ice-rich permafrost

(Jorgenson, 2013) and thus serve as vital and visual indicators of permafrost degradation. An RTS typically consists of a sub-vertical ice-rich headwall and a gentle slump floor occupied by mudflows (Ballantyne, 2018). The triggering factors and mechanisms include coastal erosion, high air temperatures, extreme precipitation, and human disturbance (Balser et al., 2014; French, 2018; Niu et al., 2005). Once initiated, ablation of the exposed ice-rich permafrost leads to the upslope retreat of the headwall at a rapid rate and disruption of vegetation cover. RTSs can significantly disrupt the local environment, for instance,

causing damage to infrastructure (Hjort, 2022), changing ecosystems (Kokelj and Jorgenson, 2013), and triggering the release of carbon previously stored in the frozen ground (Turetsky et al., 2020).

Compared with the counterparts in the circum-Arctic, there still lacks basic knowledge of RTSs locations on the Qinghai-Tibet Plateau (Mu et al., 2020), with only limited studies identifying RTSs in subregions of the QTEC. For instance, Niu et al. (2016) identified 42 slope failures (some of them are RTSs) by manually interpreting SPOT-5 imagery and field investigations within

a 10 km lateral zone along the Qinghai-Tibet Highway from Wudaoliang to the Fenghuo Mountain pass. Luo et al. (2019) manually interpreted 438 RTSs using a series of satellite images from 2008 to 2017 covering the Beiluhe region. None of the previous works obtained a comprehensive RTS inventory for this vital area due to the challenges of visiting RTSs in the remote and harsh permafrost regions or mapping them from remote sensing imagery (Huang et al., 2020).

Several methods have been used in mapping RTSs in a large area, including manual delineation and automatic recognition.

Lewkowicz and Way (2019) used the Google Earth Engine Time-lapse dataset to visually locate and delineate terrain changes on Banks Island in the Canadian Arctic. However, manual delineation is time-consuming and has a chance of missing possible RTSs. Deep learning techniques automate several fields, such as identifying targets and classifying various land covers in

remote sensing images. For permafrost-related landforms identification, Zhang et al. (2018) used Mask R-CNN to delineate ice-wedge polygons in high-resolution aerial images covering northern Alaska. Abolt and Young (2020) used deep learning and 50-cm-resolution DEMs to identify ice-wedge polygons near Prudhoe Bay, Alaska. Nitze et al. (2021) tested the regional transferability and potential for the deep learning approach in inferring RTSs in the pan-Arctic. These studies proved the applicability of deep learning in mapping permafrost-related landforms in remote sensing images and emphasized the importance of the quality and quantity of the training dataset. However, many cryospheric studies, this one included, lacked label data that are readily used in training. Set against this background, we identified and delineated RTSs along the whole QTEC by combining the efficiency of the deep learning model in mapping with the reliability of human input based on the deep-learning-based mapping method proposed by Huang et al. (2020).

This study aims to obtain a comprehensive inventory of RTSs with high accuracy along the QTEC using a semi-automatic method and plenty of supplementary datasets. Apart from this, using the topographic, soil properties, vegetation data, we reveal the spatial distribution characteristics of RTSs.

## 2    Study Area

The study area is the permafrost region along the Qinghai-Tibet Engineering Corridor, defined based on the maps of Tong et al. (2011) and Zou et al. (2017). The study area (Figure 1a) has a length of ~550 km along the Qinghai-Tibet Railway and Highway and a total area of ~54,000 $km^2$ (lying within the coordinates 90.91° E to 95.15° E, and 31.74° N to 35.99° N, Figure 1b). The mean annual ground temperature on the natural ground is -4−0°C (Jin et al., 2008; Wu and Zhang, 2008; Wu et al., 2012). Around half of the permafrost in the region has relatively high ground ice content with a thin active layer (Cheng, 2005; Yang et al., 2010). In many locations, the surface vegetation cover has been destroyed or removed because of anthropogenic and animal activities, which expose the bare ground to the air and increase the instability of this region (Jin et al., 2008; Wu et al., 2012). Thermokarst landforms, including retrogressive thaw slumps, thermo-erosion gullies, and thermokarst lakes, are widely distributed across the Qinghai-Tibetan Plateau (Huang et al., 2018; Mu et al., 2020; Niu et al., 2012). Some RTSs developed in the area are perilously close to the line of the Qinghai-Tibet Highway (Niu et al., 2005), as Fig. 1c shows a typical example.

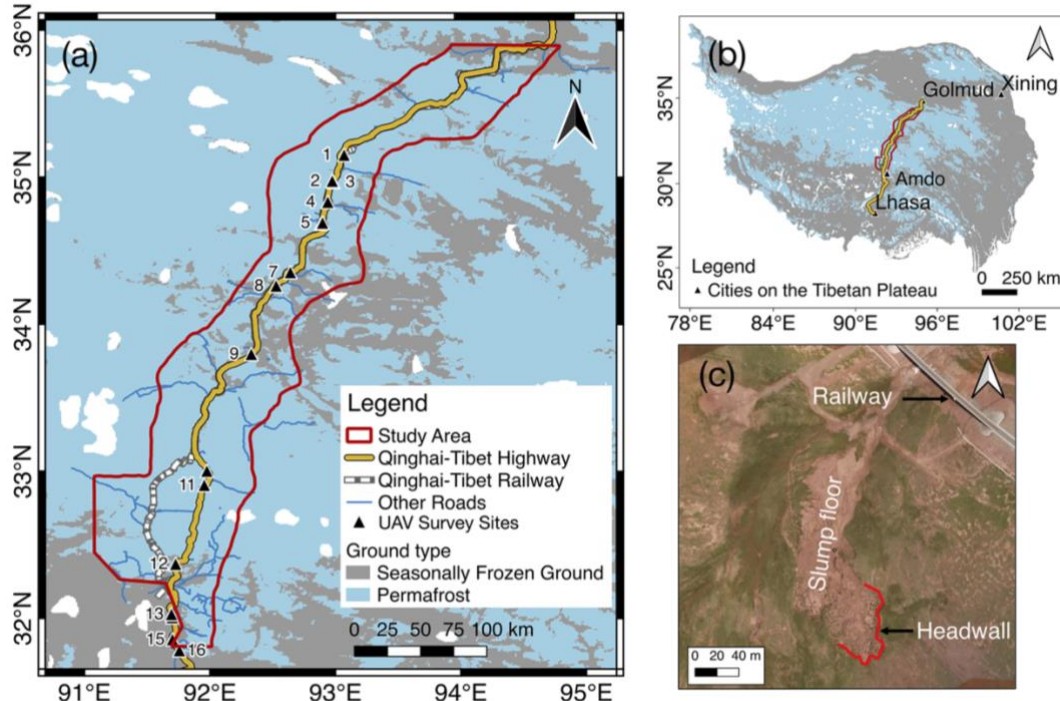

**Figure 1. (a) Coverage of the study area and the permafrost distribution. The red boundary is the extent of the study area. The yellow line is the Qinghai-Tibet Highway, and the diced line is the Qinghai-Tibet Railway, most of which runs close to the highway. Blue lines represent other national roads. The background is the permafrost distribution map produced by Zou et al. (2017), with white patches representing lakes or glaciers. The black triangles label the sites where we conducted UAV investigations. (b) The location of the study area on the Qinghai-Tibet Plateau. (c) A UAV photo of an RTS near the Qinghai-Tibet railway (center location: 92.883° E, 34.709° N).**

## 3   Data sources

We collected PlanetScope Scenes (Planet Team, 2017) with a spatial resolution of 3 meters acquired in July and August during the years 2016 to 2020. In addition to the multi-year PlanetScope images, the following supplementary data were used for reference in manual inspection: Landsat-5 and 8, Sentinel-2, unmanned aerial vehicle (UAV) images, the "World Imagery" provided by Esri, and the digital elevation model (DEM) from the Shuttle Radar Topography Mission (SRTM) (Farr et al., 2007). We downloaded Landsat and Sentinel-2 images taken before 2016 through the Google Earth Engine (Gorelick et al.,

2017). Landsat-5 carried with sensor Thematic Mapper provides images with 30 m visible bands. Landsat-8 used the Operational Land Imager sensor to obtain images with resolutions of 30 m for visible bands and 15 m for the panchromatic band. Sentinel-2 has achieved images since 2015 and provides images with a resolution of 10 m for the red, green, and blue bands. We used the flying platform DJI P4 Multispectral to obtain the UAV images with around 15-cm resolution in 16 near

roads sites where 23 RTSs candidates are located. We also accessed the high resolution (< 1 m) satellite imagery via Esri
Wayback Imagery (Esri Inc., 2018), which archived all published versions of World Imagery. Moreover, we calculated the slopes and aspects using the 30-m DEM.

To further analyze the RTSs distribution patterns and associated environmental factors, we used topo-climatic, hydrological, vegetation, and soil datasets, including (1) the annual potential incoming solar radiation (PISR), calculated using the method described by Kumar et al. (1997), (2) the stream networks simulated by SAGA GIS based on the DEM, (3) vegetation types
(data source: Wang et al., 2016); and (4) soil textures (data source: Food and Agriculture Organization of the United Nations, 2019). All the data are listed in Table 1.

**Table 1 List of the data used for mapping RTSs and analyzing their spatial distribution.**

| | Acquisition time | Spatial coverage | Spectral bands | Spatial resolution | Purpose | Source/Reference |
|---|---|---|---|---|---|---|
| PlanetScope Scenes | July, August 2019 | QTEC | red, green, blue | 3–5 m | Automatically delineating | Planet Team, 2017 |
| | July and August during the years 2016 to 2020 | | | | Manual inspection | |
| LandSat-8 | 2013–2016 | RTS locations and the surrounding areas within 1km | Panchromatic band | 15 m | Manual inspection | Google Earth Engine |
| | | | red, green, blue | 30 m | | |
| LandSat-5 | 2009–2016 | | red, green, blue | 30 m | | |
| Sentinel-2 | 2015–2016 | | red, green, blue | 10 m | | |
| UAV images | August 2020; July 2021 | 16 Selected sites along the Qinghai-Tibet Highway | red, green, blue | ~ 15 cm | Manual inspection | Field surveys |
| ESRI World Imagery | Since 2010 | QTEC | / | < 1 m | Manual inspection | Esri Inc., 2018 |
| SRTM DEM | 2000 | QTEC | / | 30 m | Manual inspection and analyzing RTS distribution patterns | Farr et al., 2007 |

| Vegetation type | / | QTEC | / | 1 km | Analyzing RTS distribution patterns | Wang et al., 2016 |
|---|---|---|---|---|---|---|
| Soil textures | 2010 | QTEC | / | 1 km | Analyzing RTS distribution patterns | Food and Agriculture Organization of the United Nations, 2019 |

## 4     Methodology

### 4.1     Pre-processing of PlanetScope images


We built an automated pipeline to download and pre-process the PlanetScope images (Huang et al., 2018), including extracting RGB bands to composite natural-color images, converting them from 16-bit to 8-bit using a linear transformation, tiling and mosaicking them to cover the entire study region. We used the processed images in 2019 to train the deep learning model and infer RTSs and used images from the other years for manual inspection.

### 4.2     Iterative mapping of RTSs


We applied a deep learning architecture called DeepLabv3+ (github.com/tensorflow/models/tree/master/research/deeplab) to identify possible RTSs, and determined RTSs from these potential candidates based on human knowledge and supplementary datasets. The DeepLabv3+ model (http://download.tensorflow.org/models/deeplabv3_xception_2018_01_04.tar.gz) we used was pre-trained using the ImageNet dataset (Russakovsky et al., 2015), making the model parameters effective in extracting general image features. To make the model feasible for identifying RTSs, we copied the architecture and parameters of the pre-trained model and fine-tuned all theparameters using corresponding PlanetScope images and labels as training data. Because the initial training data were derived from the work of Huang et al. (2021) and only included 300 RTSs in the Beiluhe region, they were insufficient for fine-tuning the deep learning model and would have led to inferior results containing multiple misidentifications and missing some RTSs. To overcome this problem and obtain a complete inventory, we adapted an iterative mapping strategy using optimized training data.



The flowchart of the method is illustrated in Fig. 2. The main steps were (1) collecting training polygons and preparing training data (Section 2.2); (2) training and fine-tuning the neural network DeepLabv3+; (3) predicting RTSs in the whole region using the 2019 PlanetScope images and reserving newly inferred polygons; (4) manually inspecting 2016–2020 time-lapse images of each new polygon to determine RTS boundaries;  and (5) adding the newly found RTSs into the positive training dataset, and optionally adding limited polygons covering representative misidentified RTSs into the negative training dataset. Then we repeated steps (2) – (5) until no new RTSs were found. Facing the difficulty due to lacking of training polygons, these iterative


experiments succeeded in obtaining a more comprehensive and representative training dataset by adding newly identified RTSs and a small number of non-RTS polygons in the next iteration. Further details are provided below.

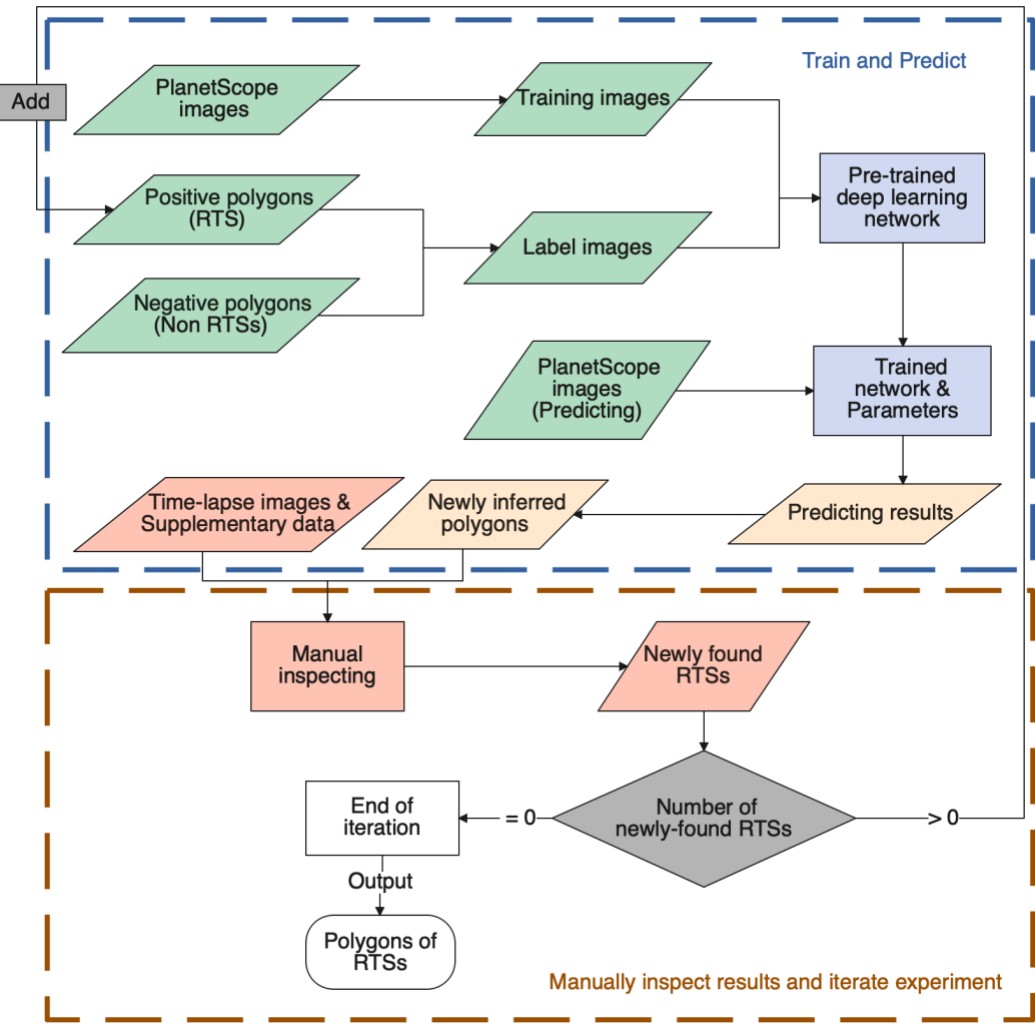

**Figure 2. Workflow of the deep-learning-aided semi-automatic method.**

In every iteration, we trained the model using the confirmed RTS polygons and negative training data of representative non-RTS polygons together with the PlanetScope images. The details of preparing training images and label images are given in Huang et al. (2018). After every iteration, we manually inspected the newly inferred polygons using the 2019 PlanetScope images, time-lapse images, as well as other supplementary data listed in Section 2.2. To prepare the time-lapse images, we first extracted sub-images from PlanetScope images collected in 2016–2020 based on the bounding boxes of deep-learning-inferred polygons with a buffer size of 300 m. We then used these chronological sub-images to make time-lapse images, with which we could visually inspect the temporal changes of RTSs. The manual inspection was based on the geomorphic features of the RTSs and their annual changes. We manually identified the headwalls based on the annual RTS retreating direction and

direction of uphill and set four criteria for improving inspection accuracy: (1) the headwall must be located at the highest

elevation inside an RTS, (2) RTSs present a yellowish-brown color in the images because of vegetation cover degradation and bare ground emergence, (3) the headwalls must be arcuate and nearly vertical, thus tend to be partially covered by narrow bands of shadows, and (4) the active RTSs retreat in an upslope direction at a rapid rate, and their retreat can be identified in the time-lapse images. One example of an RTS is shown in Fig. 3, together with the criteria we identified in the image. Then, for some inaccurate polygons, we manually modified the boundaries (e.g., Fig S1). Limited by the image resolution of 3 meters,

we need at least ~55 pixels to identify the features of thaw slumps, so the minimum mapping unit (MMU) we set is 0.05 ha. In the case of several RTSs that were stable in 2016-2020, we used multi-source images to extend the time span. One example of RTS shown in Fig. 4 was larger in 2013 than it was in 2010, but its area remained almost the same in subsequent years. For those near roads polygons that were easy to approach, we went to the field and collected UAV images, allowing us to further improve the reliability of the mapping results (e.g., Fig. 1c). Two experts manually inspected the results independently, costing

2 to 6 hours per iteration. The numbers of training polygons, deep-learning-inferred polygons, and newly identified RTSs in each iteration are listed in Table 2. To identify RTSs that are near roads, we measured the distance between the geometric center of an RTS and the roads. Considering the sizes of RTSs and their fast retreat rates, which can sometimes reach 212 m per year (Huang et al., 2021), we set the distance threshold as 500 m. Using the time-lapse images (data are available from Xia et al., 2021), we further subdivided RTSs into four groups: RTSs initiated before July or August 2016, 2016 – 2017, 2017–

2018, and 2018 – 2019 (between two summers).

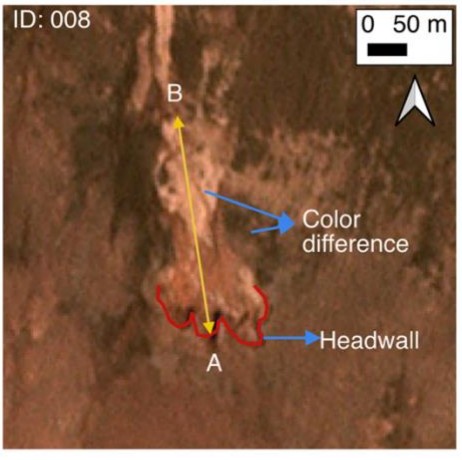
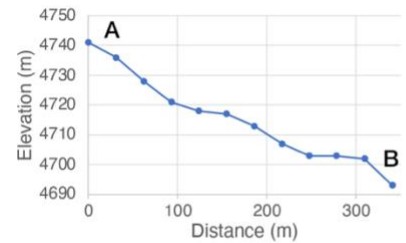

(1) The headwall has a higher elevation in an RTS

(2) Color difference compared to the surroundings.

(3) Arcuate and shaded headwall.

(4) The RTS retreat in an upslope direction over time.

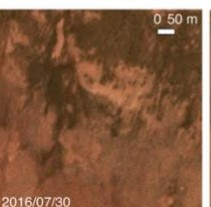
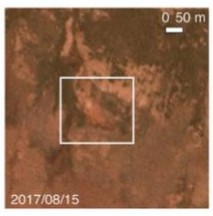
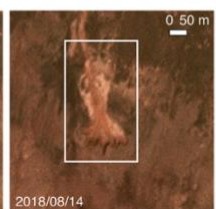
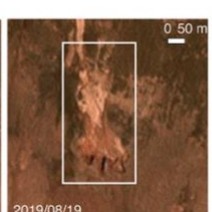
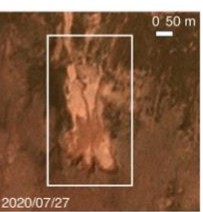

**Figure 3. An example of an RTS shown in the PlanetScope image with corresponding criteria illustrated for manual checking. The ID was assigned by us in the inventory. The white polygons highlight the RTS in the images, and this RTS is initialized after August 2016. Basemap data © Planet Labs Inc.**

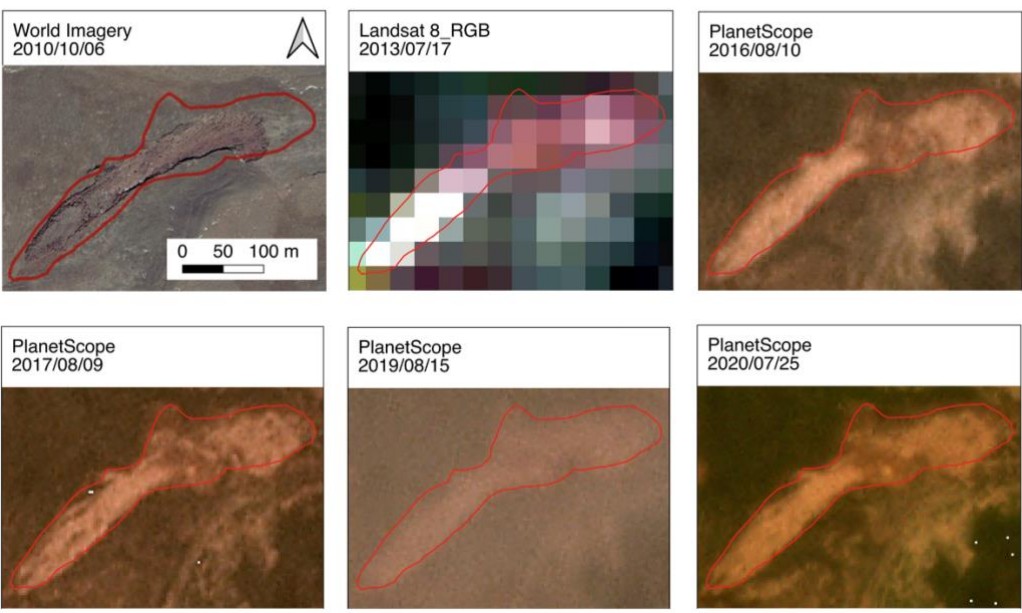

**Figure 4. Temporal images of an RTS from various remote sensing data sources, including PlanetScope images (© Planet Labs Inc), Landsat images, and World Imagery (Esri Inc., 2018). The image from World Imagery cannot be downloaded, so it was a screenshot without a scale. The ID of this RTS is 166. The red polygons represent the boundaries of the RTS, based on the 2019 PlanetScope images.**

**Table 2. Summary of iterative mapping. The positive polygons are RTSs boundaries. The negative polygons outline some non-RTS landforms or landcover that appear similar to RTSs in the PlanetScope images. The deep-learning-inferred polygons are the output of the DeepLabv3+. Newly found RTSs are polygons manually selected from the deep-learning-inferred polygons. We recorded the total number of RTSs for every iteration in the 'number of RTSs'.**

| Iteration number | Training | | Prediction | Manual inspection | Number of RTSs |
|---|---|---|---|---|---|
| | Positive polygons | Negative polygons | Deep-learning-inferred polygons | Newly found RTSs | |
| 1 | 300 | 72 | 2064 | 149 | 449 |
| 2 | 449 | 72 | 2842 | 196 | 645 |
| 3 | 645 | 78 | 3153 | 73 | 718 |
| 4 | 718 | 78 | 10510 | 86 | 804 |
| 5 | 804 | 90 | 4609 | 34 | 838 |
| 6 | 838 | 90 | 3362 | 4 | 842 |
| 7 | 842 | 90 | 5033 | 21 | 863 |
| 8 | 863 | 90 | 3622 | 12 | 875 |

| 9 | 875 | 90 | 4031 | 0 | 875 |

## 4.3 Uncertainty assessment for RTS inventory

We manually assigned a probability for each mapped RTS as an uncertainty indicator based on the availability of multi-temporal remote sensing imagery and coverage of field validation. Due to the lack of ground truth in the entire QTEC, we cannot quantify the accuracy of the whole inventory. Considering the lack of field evidence for each RTS, and the drawbacks of remote sensing imagery, such as indirect observation and limited spatial resolution, we assigned low or medium probability for an RTS that does not strictly meet the four criteria in the manual inspection. For instance, the ones that retreated abruptly in one year but were stable in other years, or their changes were too subtle to identify. The numbers of the RTSs with high, medium or low probability are 810 (92%), 33 (4%), and 32 (4%), respectively.

## 5 Results

The inventory we compiled includes 875 RTSs along the Qinghai-Tibet Engineering Corridor (Figure 5). The largest RTS has an area of 24.03 ha and the smallest one is 0.05 ha; whereas 98.5% of them are smaller than 10 ha (Figure 6a). Together they affect 1700 ha land on a 5,400,000 ha study region. Altitudes in this whole study area vary from ~3300 m to ~6200 m. Around 90% of the RTSs were found at medium elevations (4582–5010 m), and the highest was at an elevation of 5394 m (Figure 6b). The RTSs tend to be located on north-facing slopes with gentle gradients ranging from 1.2° to 18.1° (Figure 6c and 6d). Most of them (67%) are located on slopes with gradients of 4–8°. They also tend to occur in areas where the annual PISR ranges from 2900 to 3200 kWh m$^{-2}$, while the entire study region potentially receives solar radiation from 2500 to 3450 kWh m$^{-2}$ (Figure 6e). We also found 209 RTSs adjacent to the simulated stream networks. The main vegetation types in the study region are swamp meadow, alpine meadow, alpine steppe, and arid desert meadow. The alpine meadow areas contain ~75% of the RTSs (Figure 6f). Soil texture analysis indicates that a large portion of the surface soil is loam and sandy loam (~23.3% and ~71.3%, respectively), only 5.4% is clay, sandy clay loam, sand. Strikingly, ~55% of the RTSs are in areas covered by loam (Figure 6g). These heterogeneities illustrate the development of RTSs needs specific environments, such as regions with massive ground ice and sloped terrains, thus limiting RTSs to regional clusters. Our inventory revealed that ~50% of the RTSs are densely clustered in the west of the Beiluhe region (e.g., Figure 5b), while the others are sparsely scattered across the other subregions (Figure 5a). The lack of uniformity in their distribution is further shown by the density maps of the total affected area in 10 km × 10 km grid cells (Figure 7a).

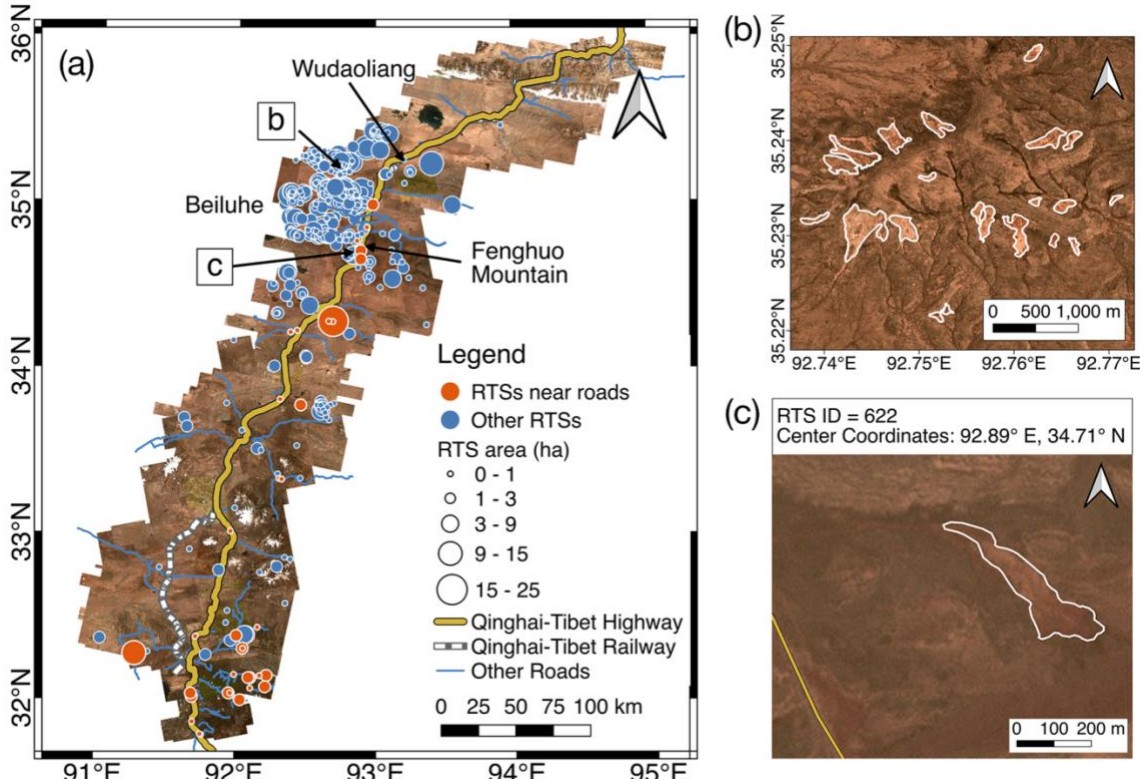

**Figure 5. (a) The map of the 875 delineated RTSs. The circle sizes indicate the RTSs' area. Orange circles are RTSs close to roads, while blue circles show other RTSs. (b) Examples of the delineated RTSs in the Beiluhe region, with the white polygons representing the boundaries of RTSs. (c) An example of an RTS adjacent to the Qinghai-Tibet Highway (yellow line). Basemap images © Planet Labs Inc.**

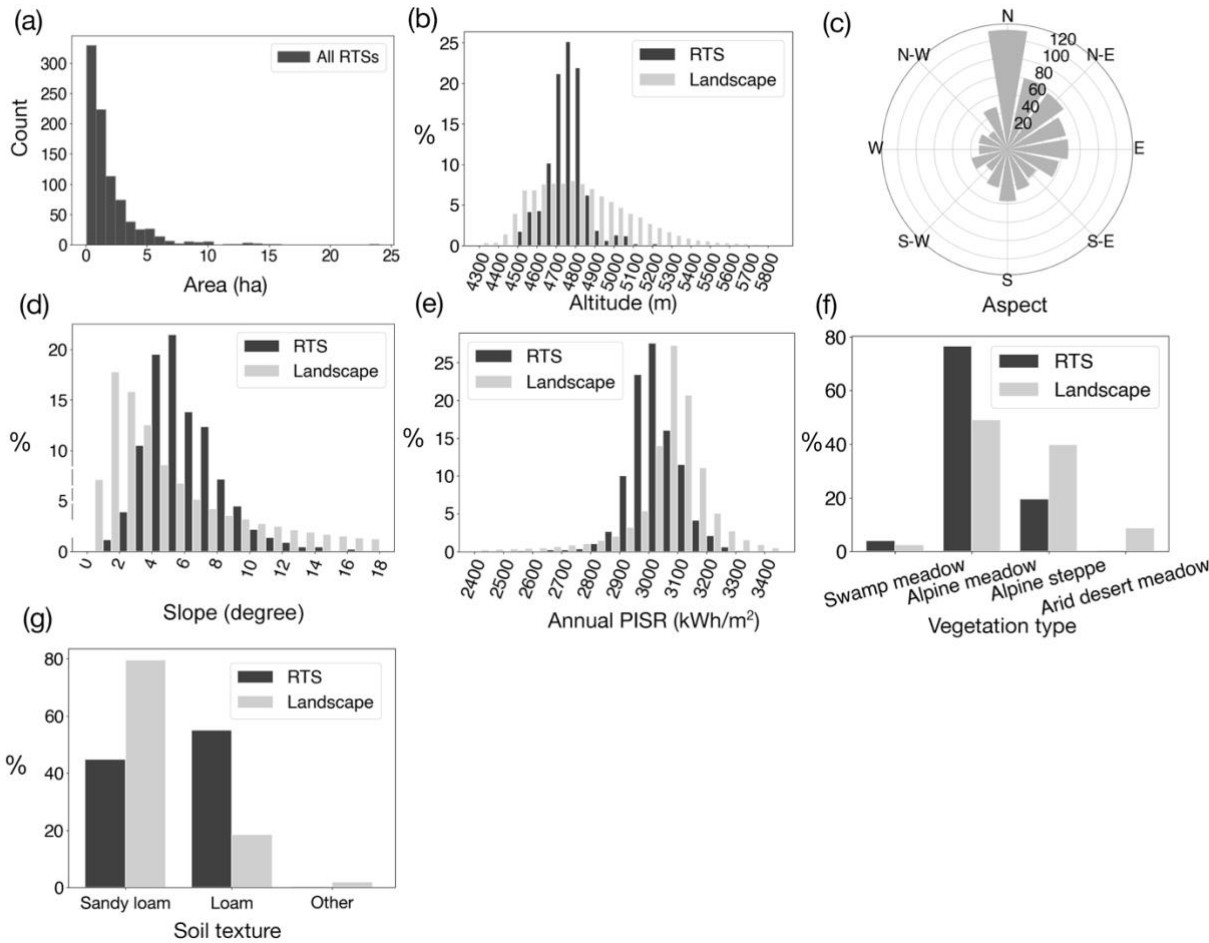

**Figure 6. Statistical summaries of the RTSs' geometric features and terrain properties. (a) The histogram shows the area of all the RTSs in the research region. (b) The elevation frequency of the landscape and RTSs. Landscape means the entire study region. (c) The slope aspects of RTSs, with radial axis representing the number of RTSs. (d) The slope frequencies of the landscape and RTSs. (e) The annual PISR frequencies of the landscape and RTSs. (f) The vegetation type distribution of the landscape and RTSs. (g) The soil texture distribution of the landscape and RTSs.**

We further identified 38 RTSs that are close to roads. Figure 5c presents an example whose center is ~400 m from the highway. The RTSs near roads are moderate in size, with an average area of 0.97 ha, and around 86.8% of the RTSs are smaller than 2 ha. The largest one has an area of 24.03 ha and is near the Yaxi Co lake. The smallest one has an area of only 0.128 ha.

Our temporal analysis revealed that there were 306 RTSs before July or August 2016. From summer 2016 to summer 2017, 455 new RTSs emerged, constituting more than half of the overall number of RTSs included in the inventory. Only 21 and 55 RTSs formed during 2017–2018 and 2018–2019, respectively. From the distribution map showing the initiating years of RTSs

in grid cells of 25 km × 25 km (Figure 7b), we observed that many of the newly initiated RTSs are located in the west of the Beiluhe region.

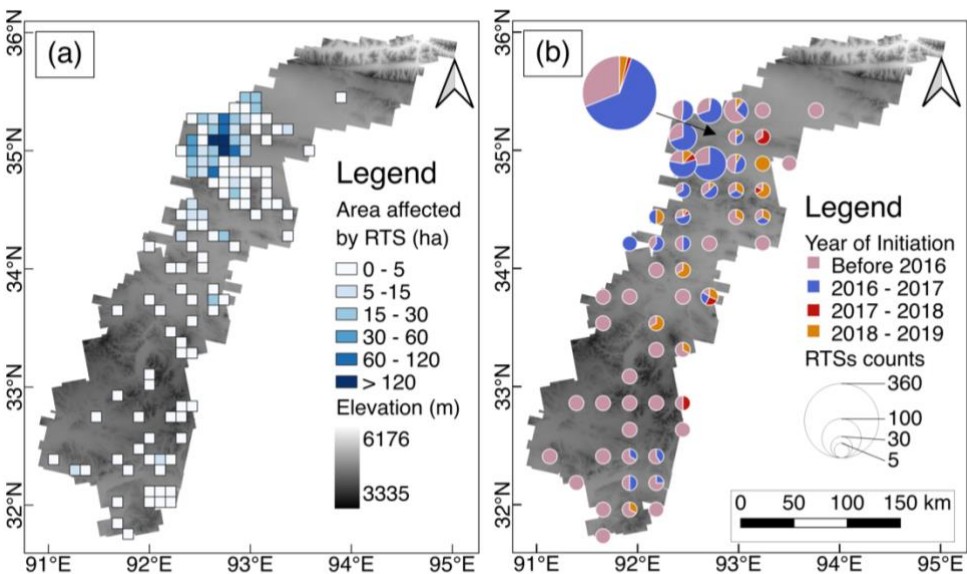

**Figure 7. (a) Areas affected by RTSs in grid cells of 10 km × 10 km. (b) The distribution map of RTSs with different initiating years in grid cells. For clear visualization, we set the cell size as 25 km × 25 km in (b). The background is a map elevation based on the Shuttle Radar Topography Mission DEM (Farr et al., 2007).**

## 6    Discussion

### 6.1    Possible controlling factors of RTS spatial distributions

Most of the RTSs are in the western part of the Beiluhe region, and a small portion of them are sparsely distributed along the roads. The uneven distribution may be controlled by topographic factors (Wang and French, 1994), hydrological factors, soil texture, vegetation, and human activities. (1) Proven by the statistical analysis of topographic features of RTSs, as the majority of the RTSs are in the Beiluhe region, the RTSs in this clustered region dominate the distribution characteristics along the QTEC. In other words, RTSs prefer to occur on gentle north-facing slopes, at medium elevations, and in locations receiving relatively low annual PISR. The main reason is that water tends to accumulate on gentle slopes, resulting in high soil moisture contents and decreased internal friction of the soil mass (McRoberts and Morgenstern, 1974). Moreover, the north-facing slopes with relatively low PISR have a thinner active layer than their south-facing counterparts. As a thin active layer is easier to be removed by thermokarst processes, the possibility of exposing the permafrost underneath will increase. The soil moisture content is also higher on land which receives low PISR (Lin et al., 2019). All the topographically controlled moisture availability is highly related to the formation of excess ground ice near the top of permafrost (Lin et al., 2020). (2) The ground

near streams tends to contain a higher water content. (3) The vegetation types also impact the distribution of RTSs, as we have shown that many RTSs are in alpine meadows. Since alpine meadows grow on land with more water content than alpine steppe (Yin et al., 2017), permafrost underneath may contain more ice. (4) The results show that the RTSs tend to develop on the land covered by loam. Silt fraction, which influences the frost susceptibility of the host sediment, is higher for loam than for sandy loam, and the ground consequently has a higher ice content (Gilbert et al., 2016). In sum, all these terrain factors, potentially related to the ice content, may exert a confounding influence on RTS formation. (5) We found 38 RTSs that are near roads, with only 7 of them in the Beiluhe region, a vulnerable area where 474 RTSs are located. It proves that engineering can minimize the impact that infrastructure has on permafrost. Excavation for soils and gravel during road construction damaged the vegetation cover in the 1980s, which led to the thawing of the exposed ice-rich permafrost and resulted in the initiation of many RTSs (Luo et al., 2019). Engineers began to realize that human activities accelerated permafrost degradation, and after 1980 adopted various methods to protect the permafrost (Luo et al., 2019). Moreover, the limited RTSs near roads indicate that it is possible, even in a vulnerable region, to select relatively stable ground for the construction of facilities and minimize the damage caused by permafrost degradation. As the distribution of the RTSs helps to pinpoint unstable ground, it should be possible to plan the alignment of a new highway along the QTEC to avoid such sensitive areas.

## 6.2    Comparison with other inventories

Our inventory is the first comprehensive one along the entire corridor region. Compared with the existing RTS datasets in the subregions (Niu et al., 2016; Luo et al., 2019), our inventory has advantages in its comprehensiveness, novelty, and being open source. Based on manual interpretation from SPOT-5 imagery and field investigations, Niu's results contain 42 slope failures (some are RTSs) in 2016 in a 10-km lateral zone of the Qinghai-Tibet Highway from Wudaoliang in the north to the Fenghuo Mountain pass in the south. In this same subregion, our method detected 47 RTSs in 2019, with 4 of them having low or medium probability. Luo's 2017 results contain 438 RTSs but only cover the Beiluhe region, within which our inventory found 459 RTSs in 2019. In total, our inventory obtains 875 RTSs in the entire study area, including the part where the critical transportation infrastructure is underlain by permafrost. We also labelled RTSs near roads and provided the initiation periods, areas, probabilities, and locations of RTSs. The deep learning model and multi-source and multi-temporal images were performed in tandem to provide a more accurate inventory than the results obtained from manual inspection alone.

## 6.3    Necessity and limitations of iterative mapping

Our method combines the efficiency of the deep learning neural network with the invaluable interpretative experience of experts. Manual delineating is labor-intensive and not feasible for a large area. Deep-learning-based mapping outperforms many other automated mapping methods by a large margin, although it still produced lots of false positives and missed a few RTSs, as shown in the first few iterations in Table 2. The newly found RTSs in every iteration indicate that one-time training and predicting has a high chance of missing some RTSs due to the bias between training data and the images covering the rest

of the study area. As proved by our iterative mapping (Table 2), by adding more training data, each new iteration successfully
inferred some RTSs missed in the previous mapping iterations.

The main disadvantage is that this method is still time-consuming while compared to a fully automated process. In each iteration, the deep learning model inferred 2 to 5 thousand polygons which need to be manually inspected. Another problem is that we may still miss some small RTSs and misidentify other landforms, for instance, drained ponds and artificial pits. Although we have already used multi-source images to guarantee the accuracy of the RTS polygons, the imagery resolution
limitation still exists, which restricts the MMU to 0.05 ha. Moreover, some RTSs that have re-vegetated on the surface cannot be identified using remote sensing images alone.

## 7    Conclusions

This study successfully used deep learning to infer possible retrogressive thaw slumps and temporal multi-source images to visually inspect retrogressive thaw slumps over a large area. This inventory of 875 thaw slumps fills the gaps in the RTS data
along the corridor and provides a diverse and representative training dataset for automatically delineating thaw slumps in even larger areas. Through statistical analysis of the terrain properties, we found that (1) the RTSs along the QTEC tend to develop on north-facing slopes with gentle degrees and tend to appear at medium elevations or areas receiving less solar radiation; (2) 209 RTSs are near stream networks; (3) a large portion of the RTSs are located on the ground covered with alpine meadows; (4) RTSs develop more frequently in areas covered by loam soil. The inventory of 38 RTSs that are near roads indicates the
human impact on permafrost and provides us with data to assess the ground stability while planning a new highway. The abnormal increase between 2016 and 2017 is worth further investigation. For instance, we can lengthen the time span and explore the relationship between the number of newly initiated RTSs and meteorological variables such as temperature and precipitation.  As the first attempt of mapping RTSs in the Qinghai-Tibet Engineering Corridor from high-resolution images, the results we obtained can potentially serve the policymakers and stakeholders with the information necessary to pursue
sustainable social-economic development on the Qinghai-Tibet Plateau.

## Data availability

The PlanetScope CubeSat images are copyrighted by Planet Labs Inc., restricted by commercial policies and are not open to the public. The Landsat 5/8 and Sentinel 2 images are publicly available through the U.S. Geological Survey and the European Space Agency, respectively, and can be downloaded via the Google Earth Engine. The Esri World Imagery can be accessed
via the Esri Wayback Imagery: https://livingatlas.arcgis.com/wayback/. The thaw slump inventory is accessible through Xia et al. (2021), Zenodo, https://doi.org/10.5281/zenodo.6397028. The Chinese version is in the National Tibetan Plateau/Third Pole Environment Data Center (Pan et al., 2021; Li et al., 2020), with link DOI: 10.11888/Cryos.tpdc.272672.

## Author contributions

LH and LL designed the study and received funding from the Hong Kong Research Grants Council. LH and ZX processed the
data, obtained the inventory, and wrote the manuscript. LL, LH, TZ, and FN revised the manuscript. CF and SJ contributed to
the field investigation. LH, ZL, JL, and FN provided the initial training data.

## Competing interests

The authors declare that they have no conflict of interest.

## Acknowledgements

This research was supported by the Hong Kong Research Grants Council (CUHK14303119) and CUHK Direct Grant for
Research (4053426 and 4053481), the Second Tibetan Plateau Scientific Expedition and Research (STEP) program
(2019QZKK0905), the National Natural Science Foundation of China (42071097), and the Strategic Priority Research
Program of the Chinese Academy of Sciences (XDA2010030805). We thank Jie Chen and Defu Zou from Northwest Institute
of Eco-Environment and Resources, CAS, for valuable assistance in the fieldwork in July 2021. This paper is dedicated to the
memory of Professor Tingjun Zhang.

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
