# Peer review of "Retrogressive thaw slumps along the Qinghai-Tibet Engineering Corridor: a comprehensive inventory and their distribution characteristics"

_Earth System Science Data, 2021_

## Author Comment (AC1)

**Responses to RC1**

(Reviewer's comments in black; authors' responses in blue)

This manuscript presents an important study on a comprehensive inventory of retrogressive thaw slumps (RTSs) along the Qinghai-Tibet Engineering Corridor (QTEC). An iteratively semi-automatic method with manual inspection were utilized to ensure the reliability of results, which should be very difficult to validate due to the lack of field evidence. The manuscript is well prepared, I suggest it should be a good study after addressing the following comments. Links between RTSs and geographic environment and environmental changes require further analysis to help reader understand mechanisms behind the distribution characteristics.

Thank you for the insightful and detailed comments. We addressed all the comments carefully with our point-by-point responses given below.

(1) It is suggested to provide a table list of the data and the purpose.

**Response**: Following your suggestion, we have added a table as shown below to list the data used.

Table 1 List of the data used for mapping RTSs and analyzing their spatial distribution

| | Acquisition time | Spatial coverage | Spectral bands | Spatial resolution | Purpose | Source/Reference |
|---|---|---|---|---|---|---|
| PlanetScope Scenes | July, August 2019

July and August during the years 2016 to 2020 | QTEC | red, green, blue | 3–5 m | Automatically delineating

Manual inspection | Planet Team, 2017 |
| LandSat-8 | 2013–2016 | RTS locations and the surrounding areas within 1 km | Panchromatic band | 15 m | Manual inspection | Google Earth Engine |
| | | | red, green, blue | 30 m | | |
| LandSat-5 | 2009–2016 | | red, green, blue | 30 m | | |
| Sentinel-2 | 2015–2016 | | red, green, blue | 10 m | | |
| UAV images | August 2020; July 2021 | 16 selected sites along the Qinghai-Tibet Highway | red, green, blue | ~ 15 cm | Manual inspection | Field surveys |
| ESRI World Imagery | Since 2010 | QTEC | / | < 1 m | Manual inspection | Esri Inc., 2018 |

| | | | | | | |
|---|---|---|---|---|---|---|
| SRTM DEM | 2000 | QTEC | / | 30 m | Manual inspection and analyzing RTS distribution patterns | Farr et al., 2007 |
| Vegetation type | / | QTEC | / | 1 km | Analyzing RTS distribution patterns | Wang et al., 2016 |
| Soil textures | 2010 | QTEC | / | 1 km | Analyzing RTS distribution patterns | Food and Agriculture Organization of the United Nations, 2019 |

(2) It is better to add place names such as Wudaoliang, Beiluhe in Figure 5. It is found that most RTSs are distributed over the region between Chumar River and Beilu River. Is that related to the initial training data (300 RTSs in the Beilu River basin) (10.1016/j.rse.2011.04.022)? Since it is very difficult to do a validation, is that possible to do another experiment with sparsely distributed training samples along the QTEC?

**Response**: We have added the location names: "Wudaoliang" and "Beiluhe" in Figure 5 as follows,

[Figure]

Figure 5. (a) Permafrost distribution map (Zou et al., 2017) with 875 delineated RTSs. The circle sizes indicate the RTSs' area. Orange circles are RTSs close to roads, while blue circles show other RTSs. (b) Examples of the delineated RTSs in the Beiluhe region, with the white polygons representing the boundaries of RTSs. (c) An example of an RTS adjacent to the Qinghai-Tibet Highway (yellow line). Basemap images © Planet Labs Inc.

The clustering phenomenon is not related to the initial training data. The outputs of the very first iteration (among the total nine iterations documented in the manuscript) contain thousands of polygons distributed across the Qinghai Tibet Engineering Corridor, with no prominent cluster in the Beiluhe region (Figure R1a).

Regarding the second comment, 'Since it is very difficult to do a validation, is that possible to do another experiment with sparsely distributed training samples along the QTEC', we conducted a new experiment with training data sparsely distributed along the QTEC, containing 434 randomly selected RTSs and 45 negative polygons. The network output 9192 mapped polygons (Figure R1b), among which 3777 polygons have already been inspected in our previous nine iterations, as documented in the manuscript. We manually inspected 5415 polygons and found no more RTSs. We chose not to add this new experiment to our revised manuscript because (1) the new experiment did not improve our mapping results, and (2) the experiment results of the first iteration have already proved that the RTS clusters are not related to the initial training data as we have addressed above.

[Figure]

Figure R1. (a) The results of the first iteration, in which we only used data in the Beiluhe region to train DeepLabv3+. (b) The results of the new experiment with sparsely distributed mapped polygons along the QTEC.

(3) Microwave remote sensing has complementary information to the optical images and is sensitive to the water content in soils (10.1016/j.rse.2020.111680). Sentinel-1, which is a C-band SAR since 2014, can be a good data source to identify RTSs (10.1002/2015JF003599). It is suggested to use this kind of microwave data or include them in the future work.

**Response**: We agreed that microwave remote sensing could provide complementary information to optical images and will consider including them in our future work. The relatively low spatial resolutions and the speckle noise make it is extremely challenging, if ever possible, to visualize RTSs on images. Firstly, the raw spatial resolutions of 2.7x22 m to 3.5x22 m are insufficient for identifying RTSs, as half of the RTSs have areas smaller than 100x100 m. Secondly, the Sentinel-1 data are adversely affected by speckle noise, making the RTS in the image not distinguishable from the surroundings (Figure R2). The access to very high-resolution (<5 m) SAR imagery is still limited. The similar microwave data to those in 10.1016/j.rse.2020.111680 are not sufficient to identify RTS due to their low resolution, but the derived products such as soil moisture can be used to analyze RTS distribution and controlling factors. The second paper that you mentioned (10.1002/2015JF003599) used the InSAR technique to measure thermokarst subsidence. However, InSAR observables alone cannot differentiate RTSs from other disturbances, for instance, constructions, landslides, periglacial mass movements, wildfires and other thermokarst landforms. Nonetheless, it is still meaningful to explore how to utilize SAR data in the future due to its availability and complementary information.

[Figure]

Figure R2. An example of an RTS in a typical SAR image. The background is the IW product of the Sentinel-1A image, with the red polygon representing the boundary of one RTS we obtained in this study.

(4) It is very interesting to further discuss why the RTSs are concentrated in the Beilu River region. The authors have mentioned several influencing factors including topography, hydrology, soil properties, vegetation cover and human activities. It is suggested to number these outlined contents. The RTSs are one of the major components of freeze-thaw erosion and should be related to the water and heat dynamics of permafrost (10.1002/2013JF002930). Therefore, its occurrence might be correlated with the number of freeze-thaw cycles (10.1002/hyp.7930) and the phase changed water content (10.1109/TGRS.2010.2051158). From your discussions, it is still not very clear why the RTSs are concentrated in the Beilu River region. A deeper analysis with controlling factors might be needed rather than a documentation presented here.

**Response**: We have numbered the contents according to your suggestion and merged two paragraphs explaining terrain and human-induced factors into one.

The excellent papers you shared may refer to the hydrological and thermal processes of the active layer. However, RTS development is mainly caused by the abrupt thawing of ice-rich permafrost, which lies below the active layer (typically, more than two meters below the surface on the Tibetan Plateau). To the best of our understanding, although most RTSs initiation is related to the accumulation of liquid water under the active layer during the thawing season, the hydrological and thermal dynamics of the active layer derived from microwave data have limited direct impacts on the RTS distribution.

To provide geographical analyses of the distribution patterns of RTSs, we collected topo-climatic, hydrological, vegetation, and soil datasets to analyze the factors that control the distribution of RTSs, as documented in the manuscript (section 6.1). The reasons for the presence of the cluster in the Beiluhe region are worth investigating but beyond the scope of this study. We plan to obtain an RTS inventory over a larger area in the future. A larger and more comprehensive RTS inventory will enable robust and meaningful factor analyses.

---

## Author Comment (AC2)

**Responses to RC2 (Ingmar Nitze)**

(Reviewer's comments in black; authors' responses in blue)

**Summary and general comments**

The manuscript/preprint "Retrogressive thaw slumps along the Qinghai-Tibet Engineering Corridor: a comprehensive inventory and their distribution characteristics" by Xia et al., describes a geospatial vector dataset of retrogressive thaw slumps along the QT Engineering corridor. The dataset is an important piece in the scope of a recently started effort to create pan-arctic/global datasets and inventories of retrogressive thaw slumps (IPA action group) for the training and validation of machine/deep-learning models.

The manuscript describes the data collection, data processing and final dataset thoroughly, however with some points that can be improved. In addition to the description of the dataset and its creation, the authors present a further analysis of the dataset and its relation to geographical parameters.

The manuscript has a good quality. However, I see some potential for improvement in the language. Although I am not a native English speaker, I noticed that some paragraphs did have some language issues. Therefore, I would recommend to put some emphasis on language editing.

Please find specific comments regarding the dataset and manuscript below.

**Response**: Thank you, Dr. Ingmar Nitze, for your insightful, thorough, and constructive comments and detailed suggestions. We revised our manuscript accordingly as follows.

**Data**

The data are easily accessible through the PANGAEA data archive and are citable with a DOI. The authors are using the standard "ESRI Shapefile" format. Although this is the quasi standard, this data format has its drawbacks limiting the attribute name length or having multiple files. As this is a comparably small dataset, the authors may provide data also in OGC compatible "GeoJSON" or the more robust "geopackage/GPKG" format, which is a bit more flexible and often a bit easier to use. However, this is just a minor/optional suggestion.

**Response**: Thanks for the suggestion. We will use "geopackage/GPKG" as the format for the updated and our future datasets. As the Pangaea does not support "geopackage/GPKG" format, we uploaded the updated datasets to Zenodo (https://doi.org/10.5281/zenodo.6397028).

**Detailed questions and comments**

**Dataset**

If you have the possibility to make edits to the dataset, I would be happy if you could check the following suggestions. I understand that updating a published dataset is maybe not the most straightforward task.

"Probabilit": This manually assigned (kind of arbitrary) attribute is not very intuitive to me. I would rather understand it as a calculated output from e.g. the DL model. This is not a major point, but you may find a better naming. However, if not that is also alright.

**Response**: We agreed with the reviewer that 'probability' might not be suitable here to represent the meaning of how much we are sure about the mapped retrogressive thaw slumps. In the revised dataset, we explained the "Probabilit" in the metadata. We also changed the value to 'high', 'medium', and 'low' according to four criteria in the manuscript. As mentioned previously, we published the revised datasets to Zenodo.

"Source Image": PlanetScope Scene or OrthoTile?

**Response**: PlanetScope Scene. We modified the metadata and manuscript accordingly.

Year-Month: I think it might be better to split this attribute into (1) Year, (2) Months. However, tracing back to the original image, before mosaicking, would be even better. If this is possible just use some standard data format, e.g. ISO format or YYYY-MM-DD, and add the original filename. If that is not possible, just leave as is.

**Response**: Thanks for the suggestion, but it is difficult for us to trace back to the original image because we created cloud-free mosaics using PlanetScope images acquired in July and August 2019 before DL-based mapping, assuming that the change of an RTS boundary is negligible on images. We will trace image acquisition dates in our future mapping tasks.

Do you have DL model versions? This information may help with reproducibility.

**Response**: We used DeepLabv3+ to obtain the results. We added the information in the metadata. We also added the GitHub link of the model version and the link of the pre-trained model in the manuscript.

**Manuscript**

l.35 – remove "normally"

**Response**: Revised.

l.66 – "lack" typo?

**Response**: We changed it to "lacked".

l.68 – "combing" typo?

**Response**: Changed it to "combining".

l.79ff – is the vegetation cover destruction/disturbance evenly distributed or limited to certain areas? Is that somehow linked to the presence of RTS?

**Response**: The vegetation cover disturbance is unevenly distributed as the disturbance is caused by various reasons, such as human activities and thermokarst landforms destruction. We don't have a map of the vegetation cover disturbance, so the relationship with the presence of RTSs is undiscovered. We postulated potential links between vegetation type and RTS presence, then conducted a correlation analysis between them, and found that RTSs tend to develop on the land covered by alpine meadows.

L85: Figure 1a. What are the white spots? Glaciers?

**Response**: The white spots mean lakes or glaciers. We added the explanation in the caption of Figure 1.

Figure 1a: Perhaps you could use a different projection as EPSG:4326 often creates some distortions (squeezes Latitude). This is just a little suggestion, perhaps at ~30°N it is not too bad compared to high latitudes.

**Response**: We changed the projection to a widely used projection 'EPSG:102025', of which the full name is 'Asia North Albers Equal Area Conic'. The latitudinal distortion of this projection is less severe than EPSG:4326.

L92: which data product (Scenes or Orthotiles)?

**Response**: The data are PlanetScope Scenes. We have updated the manuscript accordingly.

L95: Which DEM did you use? Absolute elevations?

**Response**: The DEM we used is from the Shuttle Radar Topography Mission, which used EGM96 (Earth Gravitational Model 1996) as the vertical datum. We mentioned it at L102 in the original manuscript. To make it easy to read, we have moved it to the front in the new version. The revised sentence is

*"In addition to the multi-year PlanetScope images, the following supplementary data were used for reference in manual inspection: Landsat-5 and 8, Sentinel-2, unmanned aerial vehicle (UAV) images, the "World Imagery" provided by Esri, and the digital elevation model (DEM) from the Shuttle Radar Topography Mission (SRTM) (Farr et al., 2007)."*

l100: "in several local sites". Can you be a bit more specific? How many, how much area? Are they representative?

Do you think these spots can be added to Fig1. without "overfilling" it. If it is not possible that's no problem, just a suggestion.

**Response**: We changed it to *"We used the flying platform DJI P4 Multispectral to obtain the UAV images with around 15-cm resolution in 16 near-roads sites where 23 RTS candidates are located."*

We added these sites to the figure, as shown below (new Figure 1).

[Figure]

Figure 1. (a) Coverage of the study area and the permafrost distribution. The red boundary is the extent of the study area. The yellow line is the Qinghai-Tibet Highway, and the diced line is the Qinghai-Tibet Railway, most of which runs close to the highway. Blue lines represent other national roads. The background is the permafrost distribution map produced by Zou et al. (2017), with white patches representing lakes or glaciers. The black triangles label the sites where we conducted UAV investigations. (b) The location of the study area on the Qinghai-Tibet Plateau. (c) A UAV photo of an RTS near the Qinghai-Tibet railway (center location: 92.883 °E, 34.709 °N).

110 ff: I understand that this is a data paper, where the processing has been done, but why did you only use RGB and not the NIR band, which from my experience helps a lot (at least in the Arctic)?

**Response**: The TensorFlow version of DeepLabv3+ and the corresponding pre-trained models we used only accept three bands. Huang et al. (2020) conducted the experiments by using different bands combinations to predict RTSs in the Beiluhe River Basin and concluded that the combination of red, blue, and green bands obtained the best results. Since the NIR is sensitive to vegetation type, density, and general plant health, the possible reason for the poor performance of using the NIR is that the vegetation in the study area is sparse, so the RTSs (without vegetation cover) are indistinguishable in the NIR band.

Did you test other combinations training/inference year combinations, e.g. train on 2020?

**Response**: No, we only used 2019 images for DL training and inference, and images in other years for manual validation. We agree that the multi-temporal data are suitable for testing the transferability of the deep learning model and will include this test in our future work.

115ff Paragraph 4.2. Did you try to add more information to the deep learning model? Which are the input bands, was it only RGB?

**Response**: Yes, we only inputted RGB bands to the deep learning model. The reasons are that the TensorFlow version of DeepLabv3+ and the corresponding pre-trained models we used only accept three bands, and more information such as the NIR band does not guarantee the improvement in the results (Huang et al., 2020). Also, testing the best combination of different information for the deep learning model is out of the scope of this manuscript.

134 Typo "PlanetScpoe" (Same in the flowchart, lower green parallelogram)

**Response**: We revised the manuscript.

137 What do you mean with sub-images? How did you create them, what is their size? Please provide more information what they are.

**Response**: The sizes of sub-images are the bounding boxes of every deep-learning-inferred polygon with a buffer size of 300 m. To be specific, we changed the sentence to

*"To prepare the time-lapse images, we first extracted sub-images from PlanetScope images collected in 2016–2020 based on the bounding boxes of deep-learning-inferred polygons with a buffer size of 300 m."*

139 Please change "changes yearly" to "annual changes"

**Response**: Revised.

140 Do you automatically retrieve the headwall? This part is somehow unclear to me? I assume you are inferring the footprints, is that correct? So the headwall position is intereptation, right? Then it is logical and (1) can be omitted.

**Response**: No, we manually identified the headwalls based on the annual RTS retreating direction and direction of uphill. We revised the manuscript as follows:

*"We manually identified the headwalls based on the annual RTS retreating direction and direction of uphill and set four criteria for improving inspection accuracy: (1) the headwall must be located at the highest elevation inside an RTS".*

160/Fig4. I think you can still add a scalebar to the World Imagery I think, it has the same extent, so we can safely assume the same scale. Alternatively you could use only one north arraw and scale bar for the entire figure, as it is the same for each map.

**Response**: We updated the figure with only one north arrow and one scale bar for the entire figure, as shown below.

[Figure]

Figure 4. Temporal images of an RTS from various remote sensing data sources, including PlanetScope images (© Planet Labs Inc), Landsat images, and World Imagery (Esri Inc., 2018). The image from World Imagery cannot be downloaded, so it was a screenshot without a scale. The ID of this RTS is 166. The red polygons represent the boundaries of the RTS, based on the 2019 PlanetScope images.

165/Table1: It is not clear what "negative polygons" are for training. In a binary classification/segmentation I would assume to only have positive polygons (target class) OR a raster mask with positive and negative (background) values.

**Response**: Negative polygons are "negative training polygons", which outline some non-RTS landforms or landcover that appear similar to RTSs in the PlanetScope images. Adding training data from these negative polygons is helpful to reduce false positives significantly.

We revised the caption of Table 2 to

*"Summary of iterative mapping. The positive polygons are RTSs boundaries. The negative polygons outline some non-RTS landforms or landcover that appear similar to RTSs in the PlanetScope images. The deep-learning-inferred polygons are the output of the DeepLabv3+. Newly found RTSs are polygons manually selected from the deep-learning-inferred polygons. We recorded the total number of RTSs for every iteration in the 'number of RTSs'. "*

Table1: better use "Prediction" or "Inference" instead of "Predicting" as a heading

**Response**: Revised.

166ff: How did you handle inaccurate polygons? I sometimes experience, that thaw slumps are perhaps correctly identified, but the polygon might now outline the RTS correctly. How did you handle these cases? Please provide more information.

**Response**: We manually modified the boundaries for some inaccurate polygons. To make it clear, we added the sentence in the manuscript:

*"Then, for some inaccurate polygons, we manually modified the boundaries."*

166ff: In the same sense, did you do some fine-tuning on the DL model output? E.g. I suppose the model has some kind of probability output, where a threshold (0-100%) can be set to (1) impact the number of detected RTS and (2) impact the polygon shape in the polygonization (raster to vector) process. Could you perhaps provide a little bit of insight either here, or even better in the detailed workflow description.

**Response**: No, we did not fine-tune the DL model output. In the TensorFlow version we used, the output contained two probabilities (background or slump) for each pixel. We took the class with maximum probability as DL output without setting any threshold.

175: Do you have a specific minimum mapping unit (MMU)? Is 0.05ha you MMU? If yes, please mention that.

**Response**: Yes, it is. According to your suggestion, we mentioned the minimum mapping unit in Section 4.2. The sentence is

*"Limited by the image resolution of 3 meters, we need at least ~55 pixels to identify the features of thaw slumps, so the minimum mapping unit (MMU) we set is 0.05 ha."*

176: It would be nice if you could mention the study area size again in relation to 1700ha.

**Response**: We added it.

*"Together they affect 1700 ha land on a 5,400,000 ha study region."*

182: Does soil texture correlate to excess ground ice in these regions or is it independent?

**Response**: According to Gilbert et al. (2016), the grain size distribution, particular the proportion of silt, influences the frost susceptibility of the host sediment, for example, the degree to which the soil favours the formation of segregated ice, so the soil texture correlates to the excess ground ice. We analyzed the soil texture in the Discussion section. We modified the sentence as:

*"Silt fraction, which influences the frost susceptibility of the host sediment, is higher for loam than for sandy loam, and the ground consequently has a higher ice content (Gilbert et al., 2016)".*

190ff/Fig5. The red colors are really hard to pick up on all three maps. Please check if you can find a better color with a lot more contrast to the background. I guess for colorblind people, they might be just invisible.

For panel (a) you may just want to use the background of Figure 1, as the Planet Mosaic does not add significant information.

Panel (a): Please use some slightly different styling for the location indicators of b and c, e.g. letter in a box or so. At the moment it has the same styling as the main panel descriptor (a).

**Response**: We changed the color of the RTSs near roads to orange and the color of the other RTSs to blue. We checked the figure using the Coblis – Color Blindness Simulator, and now the figure is visible for colorblind people. For panel (a), as the mosaiced Planet image can show the data we used, we keep the background unchanged. Also, we changed the styling of b and c. We used white color for panels (b) and (c) to delineate the boundaries. The figure is shown in Figure R3.

[Figure]

Figure 5. (a) Permafrost distribution map (Zou et al., 2017) with 875 delineated RTSs. The circle sizes indicate the RTSs' area. Orange circles are RTSs close to roads, while blue circles show other RTSs. (b) Examples of the delineated RTSs in the Beiluhe region, with the white polygons representing the boundaries of RTSs. (c) An example of an RTS adjacent to the Qinghai-Tibet Highway (yellow line). Basemap images © Planet Labs Inc.

194: Fig 6. I like the comparison of histrograms (b,d,e)! It clearly shows the differences to the overall area.

Fig 6c: The red numbers are very hard to read. Please check your color scheme. Perhaps you can use lighter shades of grey for the bars. Think about changing the color of the numbers. You could provide information what the numbers mean in the caption.

**Response**: We changed the color of the bars to light grey and the color of the labels to black. We added information about what the numbers mean in the caption.

Fig 6f/g From looking at the diagrams it's not immediately clear which one is inside and which one outside. I recommend to use either barcharts with or keep the round diagrams but use different signatures for the inner vs outer part and add a legend. I may go with barcharts as the proportions are easier to pick up visually.

**Response**: Thank you! We changed them to bar charts (Figure R4).

[Figure]

Figure 6. Statistical summaries of the RTSs' geometric features and terrain properties. (a) The histogram shows the area of all the RTSs in the research region. (b) The elevation frequency of the landscape and RTSs. Landscape means the entire study region. (c) The slope aspects of RTSs, with radial axis representing the numbers of RTSs. (d) The slope frequencies of the landscape and RTSs. (e) The annual PISR frequencies of the landscape and RTSs. (f) The vegetation type distribution of the landscape and RTSs. (g) The soil texture distribution of the landscape and RTSs.

201: I think it would be nice to shortly repeat the "close to road" definition for better readability.

**Response**: Revised.

Figure 7a: It would be helpful if you could normalize the values by area. E.g. percent or ha/km². Then values become comparable to other studies.

**Response**: We set the grid size as 10 km ×10 km, so the values can be compared to other studies.

241ff: this paragraph may need some more language editing

**Response**: Revised as follows:

*"Our inventory is the first comprehensive one along the entire corridor region. Compared with the existing RTS datasets in the subregions (Niu et al., 2016; Luo et al., 2019), our inventory has advantages in its comprehensiveness, novelty, and being open source. Based on manual interpretation from SPOT-5 imagery and field investigations, Niu's results contain 42 slope failures (some are RTSs) in 2016 in a 10-km lateral zone of the Qinghai-Tibet Highway from Wudaoliang in the north to the Fenghuo Mountain pass in the south. In this same subregion, our method detected 47 RTSs in 2019, with 4 of them having low or medium probability. Luo's 2017 results contain 438 RTSs but only covering the Beiluhe region, within which our inventory found 459 RTSs in 2019. In total, our inventory obtains 875 RTSs in the entire study area, including the part where the critical transportation infrastructure is underlain by permafrost. We also labelled RTSs near roads and provided the initiation periods, areas, probabilities, and locations of RTSs. The deep learning model and multi-source and multi-temporal images were performed in tandem to provide a more accurate inventory than the results obtained from manual inspection alone."*

243: better use "novelty" instead of newness

**Response**: Revised.

246 multi-time -> multi-temporal

**Response**: Revised.

254: "some false positives" I would use a bit stronger wording, as FP vastly outnumbered true positives.

**Response**: Revised.

263/264: here you mention that you have some kind of MMU, you did not state that above (see comment further above).

**Response**: Revised, as follows:

*"Limited by the image resolution of 3 meters, we need at least 55 pixels to identify the features of thaw slumps, so the minimum mapping unit (MMU) we set is 0.05 ha."*

267 ff: in this paragraph you use the terms "thaw slumps" and "retrogressive thaw slumps". Before you used the abbreviation RTS. Please be more consistent.

**Response**: We changed "thaw slumps" to "retrogressive thaw slumps".

271: "or area receives less solar radiation". That somehow does not read well.

**Response**: We changed it to "or lands receiving less solar radiation".

Reference
G.L. Gilbert, M. Kanevskiy, J.B. Murton: Recent Advances (2008–2015) in the study of ground ice and cryostratigraphy, Permafrost Periglac. Process., 27 (4), pp. 377-389, 10.1002/ppp.1912, 2016.

---

## Author Comment (AC3)

**Responses to RC3**

(Reviewer's comments in black; authors' responses in blue)

**Major comment:**
Thaw slumps are an important phenomenon of permafrost degradation and have a significant impact on engineering, ecological processes, and the carbon cycle. This paper by Xia et al. achieves mapping thaw slumps on a large scale with high precision via combining deep learning and manually inspecting. The paper is generally well organized, the objectives are clear, and the methods are also well designed, for instance, using an iterative mapping method to find more thaw slumps with limited training data and assigning a probability for each mapped uncertain thaw slump. Therefore, the results are quite robust. To date, thaw slump investigations on the QTP are still urgent, and hence I think this important dataset would potentially serve as fundamental data for understanding the impacts of thaw slumps in the warming world. I, therefore, think this paper is a nice contribution that can be published in ESSD journal after minor revisions.

**Response**: Thank you for your positive comments and detailed suggestion.

**Specific comments:**

- P2, L34: Permafrost definition is not originally described in French (2018), please see Van Everdingen, R.O. (1998)

**Response**: We added this reference according to your suggestion.

- P2, L30: This sentence doesn't seem to constitute causality. Please revise the wording and grammar.

**Response**: We changed the sentences to:

*"The potential damage to infrastructure and the carbon emission of thaw slumps motivated us to obtain an inventory of thaw slumps. We used a semi-automatic method to get 875 thaw slumps, filling the knowledge gap of thaw slump locations and providing key benchmarks for analyzing the distribution features and quantifying spatio-temporal changes."*

- P2, L35: ... of **about** $1.06 \times 10^6$ km$^2$

**Response**: Revised.

- P2, L48: Please put the references behind the corresponding content respectively, rather than putting them at the end together.

**Response**: We separated the reference according to your suggestion. We also cited a paper about the impact of retrogressive thaw slumps (RTSs) on infrastructure. The sentence is listed below.

*"RTSs can significantly disrupt the local environment, for instance, causing damage to infrastructure (Hjort, 2022), changing ecosystems (Kokelj and Jorgenson, 2013), and triggering the release of carbon previously stored in the frozen ground (Turetsky et al., 2020)."*

- P3, L66: "cryospheric studies" rather than "cryosphere studies."

**Response**: Revised.

- P3, L75: There are too many "and" in this sentence

**Response**: Thank you. We deleted one "and". Now the sentence is

*"The study area (Figure 1a) has a length of ~550 km along the Qinghai-Tibet Railway and Highway and a total area of ~54,000 km² (lying within the coordinates 90.91° E to 95.15° E and 31.74° N to 35.99° N, Figure 1b)."*

- P9, L170: Do you mean the low probability is < 100%, and the high probability is = 100%?

**Response**: Yes, it is what we meant, but it may lead to misunderstandings. Following the suggestion from Reviewer 2, we changed the previously adopted numerical values of probability to 'high', 'medium', and 'low' according to the four criteria listed in the manuscript.

P14, L264: Please change the "ha" to SI unit.

**Response**: We used "ha" because the sizes of the retrogressive thaw slumps are moderate. If using "meter" everywhere in the paper, the number will be too long to show in a figure, while using "kilometer", the value is too small. Also, "ha" is widely used in representing the area of RTSs in other papers and is an SI-accepted unit.

**Tables & Figures**

- **Summary of RS data:** Could you please re-organize the description RS datasets in Sec.3? What about adding one more table regarding to their summary info? For example, data coverage, used bands, spatial resolution, and purpose of each dataset.

**Response**: We added one table to describe the remote sensing data. The table is shown below.

Table 1 List of the data used for mapping RTSs and analyzing their spatial distribution

| | Acquisition time | Spatial coverage | Spectral bands | Spatial resolution | Purpose | Source/Reference |
|---|---|---|---|---|---|---|
| PlanetScope Scenes | July, August 2019

July and August during the years 2016 to 2020 | QTEC | red, green, blue | 3–5 m | Automatically delineating

Manual inspection | Planet Team, 2017 |

| | | | | | | |
|---|---|---|---|---|---|---|
| LandSat-8 | 2013–2016 | RTS locations and the surrounding areas within 1km | Panchromatic band | 15 m | Manual inspection | Google Earth Engine |
| | | | red, green, blue | 30 m | | |
| LandSat-5 | 2009–2016 | | red, green, blue | 30 m | | |
| Sentinel-2 | 2015–2016 | | red, green, blue | 10 m | | |
| UAV images | August 2020; July 2021 | 16 Selected sites along the Qinghai-Tibet Highway | red, green, blue | ~ 15 cm | Manual inspection | Field surveys |
| ESRI World Imagery | Since 2010 | QTEC | / | < 1 m | Manual inspection | Esri Inc., 2018 |
| SRTM DEM | 2000 | QTEC | / | 30 m | Manual inspection and analyzing RTS distribution patterns | Farr et al., 2007 |
| Vegetation type | / | QTEC | / | 1 km | Analyzing RTS distribution patterns | Wang et al., 2016 |
| Soil textures | 2010 | QTEC | / | 1 km | Analyzing RTS distribution patterns | Food and Agriculture Organization of the United Nations, 2019 |

- **Figure 1:** Could you please add the lake info here so that authors could clearly see the missing data? You could use the public land cover maps, such as the ESA CCI or TP lake inventory from TPDC.

Response: We double-checked and found that in the map product of Zou et al. (2017), the white parts represent some lakes or glaciers. We misunderstood them as missing data. We changed the manuscript accordingly.

- **Figure 6**: (f), (g): I would suggest using bar charts instead of pie charts, so that the data may be more intuitive and easier to compare. The pie charts look a little messy.

Response: We changed them to bar charts. The figure is shown below (new Figure 6).

[Figure]

Figure 6. Statistical summaries of the RTSs' geometric features and terrain properties. (a) The histogram shows the area of all the RTSs in the research region. (b) The elevation frequency of the landscape and RTSs. Landscape means the entire study region. (c) The slope aspects of RTSs, with radial axis representing the number of RTSs. (d) The slope frequencies of the landscape and RTSs. (e) The annual PISR frequencies of the landscape and RTSs. (f) The vegetation type distribution of the landscape and RTSs. (g) The soil texture distribution of the landscape and RTSs.

---

## Referee Report (RR1)

Summary

This manuscript describes an RTS dataset in the region of QTEC that has a total of 875 RTS polygons. The authors used a deep learning approach to aid the manual delineation of RTS polygons. Although this method presented in the manuscript is not the most effective and difficult to adapt, the dataset itself serves well as a developed RTS study/training dataset, which can be helpful for the training of RTS segmentation deep learning models.

Major points

-The trained DeeplabV3+ model used pre-trained weights using the ImageNet dataset. It is sometimes questioned that ImageNet dataset's context/scene is very different from remote sensing images and therefore difficult for transfer learning. And the model performance is highly dependent on how the fine-tuning is done. So, maybe it's helpful to reveal some details on how the ImageNet pre-trained model is fine-tuned, and the accuracy on the test set. Although this manuscript is a data description paper rather than a methodology paper, but as part of the data is auto-generated, it is crucial for the readers to fully understand the data generation process. Also, there's a limitation for ImageNet pre-trained model that it was trained on RGB channels, which makes its adaption to multi-bands satellite image data very difficult. In the future maybe consider the use of BigEarthNet to pre-train the model.

-It will be helpful to have a figure of a/some successfully predicted RTS by the model, and the same RTS polygon that finally delineated from the prediction. This can help the readers to understand the manual inspect process.

-For the probability assessment, it will be helpful to have a table/figure to show the percentage/number of each category, i.e. high/medium/low. To give an overview of the quality of the dataset.

Minor points

Line 182 complied – compiled?

---

## Referee Report (RR2)

line 40 - please change "get 875 thaw slumps" to  "map 875 thaw slumps"

line 391 - please change "on north-facing slopes with gentle degrees" to " on gentle north-facing slopes"

---

## Author Response (AR2)

Dear Editor:

Thank you very much for the extensive effort you have made in handling our manuscript and identifying reviewers. We have revised our manuscript according to the insightful comments from the reviewers. All the comments (in black) are addressed with point-by-point responses (blue). The sentences copied from the manuscript are in italic format.

1. Line 23 and 294: with link: https://data.tpdc.ac.cn/zh-hans/disallow/50de2d4f-75e1-4bad-b316-6fb91d915a1a/. This link looks like a temporary link. Please change to a formal form.
Response: Thank you for the advice. We have changed the link to the formal form.

2. Please modify the article as the referee's suggestions:
(1)Line 40: please change "get 875 thaw slumps" to "map 875 thaw slumps"
(2)Line 391: please change " on north-facing slopes with gentle degrees" to " on gentle north-facing slopes"
Response: We have revised the manuscript accordingly.

3. Since you have put the data at the National Tibetan Plateau/Third Pole Environment Data Center, you are welcome to cite the relevant introduction papers into the articles as: https://doi.org/10.1175/BAMS-D-21-0004.1 and https://doi.org/10.1175/BAMS-D-19-0280.1
Response: We have added the citations in the Data available section. The revised sentence is:
" The Chinese version is in the National Tibetan Plateau/Third Pole Environment Data Center (Pan et al., 2021; Li et al., 2020), with link DOI:10.11888/Cryos.tpdc.272672."

4. Please modify the article according to the comments from the referees as shown in the attached file.
Response: We have revised the article accordingly.

5. Since the research is related to permafrost, it is suggested that the author may reference relevant articles in the special issue,such as https://doi.org/10.5194/essd-13-4207-2021, and https://doi.org/10.5194/essd-14-865-2022

Response: We added the citations in our manuscript, as follows:

*"Because the underlying permafrost on the plateau is characterized by shallow thickness and relatively high temperature (Ran et al., 2022; Wu and Zhang, 2008; Wu et al., 2010; Zhao et al., 2021; Zhou et al., 2000), it is vulnerable to degradation under climate warming and disturbance due to human activities."*

Best Regards,
Zhuoxuan XIA and co-authors

**Responses to reviewers**

1.  The trained DeeplabV3+ model used pre-trained weights using the ImageNet dataset. It is sometimes questioned that ImageNet dataset's context/scene is very different from remote sensing images and therefore difficult for transfer learning. And the model performance is highly dependent on how the fine-tuning is done. So, maybe it's helpful to reveal some details on how the ImageNet pre-trained model is fine-tuned, and the accuracy on the test set. Although this manuscript is a data description paper rather than a methodology paper, but as part of the data is auto-generated, it is crucial for the readers to fully understand the data generation process. Also, there's a limitation for ImageNet pre-trained model that it was trained on RGB channels, which makes its adaption to multi-bands satellite image data very difficult. In the future maybe consider the use of BigEarthNet to pre-train the model.

Response: We agree that the data in ImageNet dataset are distinct from remote sensing images, so we fine-tuned all layers of the pre-trained model using our dataset. To reveal the details of how we fine-tuned the model, we have revised the sentences as :*"The DeepLabv3+ model (http://download.tensorflow.org/models/deeplabv3_xception_2018_01_04.tar.gz) we used was pre-trained using the ImageNet dataset (Russakovsky et al., 2015), making the model parameters effective in extracting general image features. To make the model feasible for identifying RTSs, we copied the architecture and parameters of the pre-trained model and fine-tuned all the parameters using corresponding PlanetScope images and labels as training data."*

The accuracy on the test dataset is important in assessing the model performance, but was not provided in our manuscript for several reasons. Firstly, we didn't have the ground truths of RTSs in this region, so there was not a direct way to calculate the test accuracy. Secondly, our iterative mapping strategy utilized the power of deep learning to generate RTS boundaries. During our mapping iterations, RTSs missed in a previous mapping exercise could be identified in the later iterations, indicating that the test accuracies vary with iterations. Thirdly, we didn't divide the data or the study area into subsets of training, validation, and test. Still, we chose an area outside the study area (Figure R1) and conducted the accuracy assessment for the last trained model of the iterative mapping. The recall rate is 0.79, the precision is 0.95, and F1 score is 0.86. We decided

not to include some test results in the manuscript for the sake of conciseness and focusing on the generated inventory.

[Figure]

Figure R1. Location of the test area and the inferred results by our DeepLabv3+ model trained in the final iteration.

The data generation contains pre-process and post-processing. We illustrated the post-processing in the manuscript but didn't focus on the pre-processing. As our pre-processing is the same as Huang et al. (2018), we decided only to cite the paper and not repeat the technical details of pre-/post-processing steps.

Thanks for suggesting BigEarthNet for our future work, and we will use it in our deep learning work of mapping permafrost disturbance in Tibet, which potentially can overcome the limitation of RGB channels in ImageNet and be adaptable to other multi-band remote sensing imagery.

2. It will be helpful to have a figure of a/some successfully predicted RTS by the model, and the same RTS polygon that finally delineated from the prediction. This can help the readers to understand the manual inspect process.

Response: Following the suggestion, we have added a figure in the supplementary material showing the example of pixel-based classification result, DeepLabv3+ inferred polygons, and manually modified polygons. The figure is referred in the main text (Line 159) and copied as below.

[Figure]

Figure S1. Examples of DeepLabv3+ outputs and manually modified RTS boundaries. (a) Planet images. (b) Pixel-based classification results. (c) DeepLabv3+ predicted RTS polygons. (d) Manually modified RTS boundaries. The location is 92.76° E, 35.09° N.

3. For the probability assessment, it will be helpful to have a table/figure to show the percentage/number of each category, i.e. high/medium/low. To give an overview of the quality of the dataset.

Response: We briefly summarize the numbers of each category in our manuscript. The sentence is:

*"The numbers of the RTSs with high, medium or low probability are 810 (92%), 33 (4%), and 32 (4%), respectively."*

**Minor points**

1. Line 182 complied – compiled?

Response: Revised.